# CMOS-Compatible High-Performance Silicon Nanowire Array Natural Light Electronic Detection System

**DOI:** 10.3390/mi15101201

**Published:** 2024-09-27

**Authors:** Xin Chen, Jiaye Zeng, Mingbin Liu, Chilin Zheng, Xiaoyuan Wang, Chaoran Liu, Xun Yang

**Affiliations:** 1School of Electronic and Information Engineering, China West Normal University, Nanchong 637002, China; cxing202303@163.com (X.C.); 18783008545@163.com (J.Z.); liumb926@163.com (M.L.); clzheng925@163.com (C.Z.); 2Zhejiang Key Laboratory of Ecological and Environmental Big Data, Hangzhou 321001, China; wangxiaoyuan@zjemc.org.cn; 3Ministry of Education Engineering Research Center of Smart Microsensors and Microsystems, College of Electronics and Information, Hangzhou Dianzi University, Hangzhou 310018, China; liucr@hdu.edu.cn

**Keywords:** silicon nanowire arrays, natural light, highly controllable, low cost, high-volume preparation, signal detection circuits

## Abstract

In this article, we propose a novel natural light detector based on high-performance silicon nanowire (SiNW) arrays. We achieved a highly controllable and low-cost fabrication of SiNW natural light detectors by using only a conventional micromachined CMOS process. The high activity of SiNWs leads to the poor long-term stability of the SiNW device, and for this reason, we have designed a fully wrapped structure for SiNWs. SiNWs are wrapped in transparent silicon nitride and silicon oxide films, which greatly improves the long-term stability of the detector; at the same time, this structure protects the SiNWs from breakage. In addition, the SiNW arrays are regularly distributed on the top of the detector, which can quickly respond to natural light. The response time of the detector is about 0.015 s. Under the illumination of 1 W·m^−2^ light intensity, multiple SiNWs were detected together. The signal strength of the detector reached 1.82 μA, the signal-to-noise ratio was 47.6 dB, and the power consumption was only 0.91 μW. The high-intensity and highly reliable initial signal reduces the cost and complexity of the backend signal processing circuit. A low-cost and high-performance STM32 microcontroller can realize the signal processing task. Therefore, we built a high-performance SiNW natural optoelectronic detection system based on an STM32 microcontroller, which achieved the real-time detection of natural light intensity, with an accuracy of ±0.1 W·m^−2^. These excellent test performances indicate that the SiNW array natural light detector in this article meey the requirements of practicality and has broad potential for application.

## 1. Introduction

Natural light detectors are widely used in optical sensing, optical communication, environmental monitoring, biomedical imaging and wearable electronics [1,2,3,4,5]; for example, it is used for light perception and automatic adjustment in mobile phones, computers, dashboards, etc., with an annual demand of hundreds of millions of units. With the gradual broadening of the application fields and the increase in demand, more and more requirements have been put forward on the various indexes of natural light detectors, such as miniaturization, integration, low cost, low power consumption, high sensitivity and high reliability. Currently, commercial silicon photodetectors are unable to fully meet the above requirements [6,7]. Therefore, the academic community has begun to focus on nanomaterials to meet these demands. In the existing reports, the materials for natural light detectors include organic materials [8], nanowires [9,10], metal oxide semiconductors [11,12,13], and quantum dots [14], etc. Ahmad et al. made a highly sensitive photodetector by a composite of the organic materials P_3_HT and VOPcPhO [15], which possessed better light collection and charge transfer in the range of natural light, but the organic materials would decompose under prolonged exposure to light, with a short lifetime and poor long-term stability. Ahn et al. prepared germanium nanowires by chemical vapor deposition (CVD) and fabricated a single germanium nanowire natural light detector by using nano manipulation devices [16]. The device showed an extremely sensitive light response, but the nanowires prepared by CVD had difficulty achieving uniformity in size, and it was difficult to achieve high-volume and low-cost device preparation. In addition, the weak signal strength and low signal-to-noise ratio of a single nanowire detector make it difficult to fabricate matching backend signal processing circuits, and expensive test equipment is required to detect the photoresponse of the detectors. Maculan et al. successfully grew MAPbCL_3_ single crystals using the retrograde solubility behavior of hybrid perovskites and used them as sensitive materials for photodetectors [17]; this detector had an excellent repeatability response and long-term stability. However, its operating voltage is at 15 V, resulting in high power consumption, which limits its application potential. In addition, the material contains lead, which causes potential toxicity to the environment and the human body.

Silicon material can absorb light with wavelengths ranging from 300 nm to 1100 nm, and the wide spectral response range makes it highly suitable as a sensitive material for natural light detectors. For example, Gerald et al. prepared a single silicon nanowire (SiNW) by the CVD method and performed natural light response experiments [18]; the experimental result showed that the absorption efficiencies Q_abs_ could reach 449%. However, the single SiNW detector and the single germanium nanowire detector mentioned earlier have the same disadvantages, which makes it difficult to promote and apply them on a large scale. Iuliana Mihalache prepared a vertical SiNW array using the metal-assisted chemical etching (MACE) method and applied it to natural light detection [19]. The process of this method is simple and can prepare SiNWs in large quantities, but the SiNW arrays prepared by MACE are too dense, and the nanowires block each other. It would take a long time for the photoelectric signal to reach stability during photoelectric testing. On the other hand, SiNWs have strong activity and are easily reactive with external substances, which leads to changes in their various indicators. Due to the lack of protective structures, SiNWs are prone to breakage, which greatly reduces the long-term stability of the detector.

In this article, we present a novel natural light detector based on high-performance SiNW arrays. In the arrays, all nanowires are regularly distributed on the upper surface of the detector without blocking each other, so the detector’s response time is about 0.015 s; meanwhile, the detector can respond to light at different incident angles. SiNWs are highly reactive and prone to fracture, so we have designed a special fully wrapped structure to tightly enclose SiNWs; this structure greatly enhances the long-term stability of the detector. Unlike the planar structure of commercial silicon photodetectors, the morphology of SiNW is elongated, so the detector has high resistance (~MΩ) and a low operating current, with a power consumption of only 0.91 μW. In the Appendix A, we compare the morphology characteristics of SiNWs and silicon photodetectors in detail. In the array, multiple SiNWs are detected together, which enhances the signal intensity. Under the light intensity irradiation of 1 W·m^−2^, the signal intensity of the detector reaches 1.82 μA, and the signal-to-noise ratio reaches 47.6 dB. The high-intensity and highly reliable initial signal reduces the cost and complexity of the backend signal processing circuits. There is no need to create additional circuit modules such as amplification, noise reduction, filtering, etc., as the initial photoelectric signal can be processed and calculated by using only a low-cost and high-performance STM32 microcontroller. We have built a matching high-performance SiNW natural light electronic detection system based on the STM32 microcontroller, which achieves the detection of natural light intensity, and the testing accuracy reaches ±0.1 W·m^−2^. In summary, the preparation method and the performance of the SiNW array natural light detector in this article meet the requirements of low-cost and large-scale promotion and application.

## 2. Materials and Methods

The process flow for the preparation of a single SiNW device is shown in Figure 1. We use a (111) SOI wafer as the raw material. The top layer of the SOI wafer is pre-doped with boron, and the doping concentration is 5 × 10^15^ cm^−3^. At first, a silicon nitride film is grown on the (111) SOI wafer surface, and then tilted rectangular windows are formed on the film by photolithography (Figure 1a). The silicon nitride and top silicon in the rectangular windows are removed by dry etching, forming two rectangular etched grooves. Subsequently, the wafer was placed in potassium hydroxide solution for anisotropic wet corrosion process. Due to the much lower corrosion rate of the (111) crystal plane compared to other crystal planes, the two etching grooves become hexagonal corrosion grooves with (111) planes on each side, forming a tilted silicon thin-walled structure between adjacent grooves. Then, a self-limiting oxidation process is carried out. Due to the obstruction of silicon nitride at the top of the silicon wall, the oxidation rate in this area will be slower than other areas. A portion of the silicon in the middle of this area had not been oxidized, forming a silicon nanowire structure. After the contact electrodes and isolation channels are made in a specific location, the basic light detection unit is prepared. The three-dimensional model in Figure 1e shows the overall structure of a single SiNW device in detail. The device’s cross-section on the right shows the full-encapsulation structure of the SiNWs, which are encapsulated by silicon nitride and silicon oxide films. The surface of the SiNWs is chemically bonded to the two insulating materials, and the surface morphology is very stable; this full-encapsulation structure effectively ensures the long-term stability of the detector.

Compared to single SiNW devices, SiNW array devices can achieve better initial detection signals (a stronger signal strength and higher signal-to-noise ratio). We used the same process as the single SiNW device to prepare the SiNW array device (Figure 1f). The SiNWs in the array are regularly distributed on the upper surface of the device without blocking each other, and this structural design enables the detector to quickly perceive natural light, significantly improving its response time. In addition, multiple SiNWs working simultaneously can superimpose signals, resulting in stronger and more stable signals than a single SiNW detector. As shown in Figure 1g, in order to characterize the morphology of SiNW, we removed the silicon nitride film and silicon oxide film to obtain completely suspended SiNW. In the electron microscope image, it can be seen that the two ends of the SiNW are integrated with the bulk silicon, and the electrical connection of the SiNW can be achieved without additional transfer. Through the above simple preparation process, we achieved a high controllability, large-scale and low-cost preparation of SiNW array natural light detectors.

## 3. Results

### 3.1. The Construction of Optical Testing System

Figure 2a shows the SiNW array natural light sensor chip and its packaged device. The array has a total of 120 SiNWs with an area of 1.69 mm^2^ (1.3 mm × 1.3 mm). Figure 2c,d shows the experimental test system. We used a full-spectrum metal halide lamp as the light source, and we used the LX-107 optical power meter (Shenzhen Vicit Technology Co., Ltd., Shenzhen, China) as the standard equipment to calibrate the optical power density of the light source, we purchased LX-107 on the Taobao shopping website, and the manufacturer is Shenzhen Vicit Technology Co., Ltd. The photoelectric signal is collected through the Keithley 4200 (Keithley Instruments Company Beaverton, Beaverton, OR, USA) in real time; it can dynamically display and store photoelectric signals, Keithley 4200 was purchased from Keithley Instruments Company (The United States of America). The light-sensing mechanism of the SiNW detector follows the photoconductivity effect. Due to boron doping in the SiNW array, an acceptor level appears above the valence band. When the detector is exposed to natural light energy radiation, electrons in the valence band will be excited, and electrons that absorb low-energy photons will transition to the acceptor level, while electrons that absorb high-energy photons will directly transition to the conduction band (Figure 2b). The electron transition will form electron–hole pairs, thereby increasing the carrier concentration in the channel of the SiNW and reducing the resistance of the detector; at this time, the testing equipment can detect a significant current signal. In order to study the response of the detector to different wavelengths of light, we test the photo sensitivity of the detector (Figure 2e). It can be seen that our detector can respond in the visible light range (300 nm to 1100 nm), and the peak response wavelength is 920 nm.

### 3.2. The Optoelectronic Testing

First, we tested the light response of the detector. Figure 3a shows the detector’s response alternately in a dark environment and a 1 W·m^−2^ light intensity environment. The test results show that the detector exhibits excellent light response characteristics, and the test signal has good repeatability and stability. As shown in Figure 3b, we amplified the optical response signal to study the important parameters of the detector, such as response time, power consumption, signal strength and signal-to-noise ratio. Firstly, all the SiNWs in the array are regularly distributed on the upper surface of the detector, which can respond quickly to natural light. The detector’s response time to natural light is about 0.015 s, and the attenuation time is about 0.02 s. Due to its slender morphology, the SiNW detectors have the characteristics of high resistance and a low operating current. The power consumption of the detector is only 0.91 μW in the environment of 1 W·m^−2^ light intensity, the ultra-low power consumption makes the detector have broad application potential in the field of wearable devices. We have achieved a highly controllable array preparation of SiNWs. In the array, multiple SiNWs are detected together, and the signal intensity is increased to the microampere level. Under the illumination of 1 W·m^−2^ light intensity, the signal intensity of the detector reaches 1.82 μA. On the other hand, the random noise of each SiNW is not superposed during the testing process. According to the signal-to-noise ratio calculation formula below, the SiNW array detector in this article also has a high signal-to-noise ratio.
SNR = 20 × lg(A_signal_/A_noise_)(1)
where A_signal_ and A_noise_ are the signal amplitude and noise amplitude, respectively. After carrying out the calculation, we obtain the signal-to-noise ratio of the detector, which is 47.6 dB.

Then, we tested the response characteristics of the detector under different light intensities. As shown in Figure 4a, the light intensity was gradually increased from 0 W·m^−2^ to 2 W·m^−2^, with a bias voltage of 0.5 V. As the optical power gradually increases, more and more electrons transition to the conduction band and acceptor level, increasing the carrier concentration in the nanowires and leading to an increase in the photocurrent signal. In order to clarify the relationship between light intensity and photocurrent, we fitted the measured experimental data using a power law fitting formula to observe the relationship between light radiation intensity and photoresponsive current [20].
I_ph_∝P^θ^(2)
where I_ph_ represents the net photocurrent, P represents the light intensity, and θ represents the empirical value related to the photogenerated carrier composite activity, which can be used to describe the degree of the photogenerated carrier composite. As shown in Figure 4b, according to the fitted result, a value of θ is about 0.7, which is a deviation from the ideal value (θ = 1), which indicates the existence of photogenerated carrier recombination in the detector; this phenomenon may be caused by trap states between the Fermi level and the conduction band edge [21,22].

To further quantitatively evaluate the performance of the SiNW array natural light detector, the responsivity R and specific detectivity D* of the detectors can be calculated according to the following equations [23,24]:R = I_ph_/(P × S) = (I_light_ − I_dark_)/(P × S)(3)
D* = R × S^1/2^/(2 × e × I_dark_)^1/2^(4)
where I_light_, I_dark_, P, S, and e denote photocurrent, dark current, light intensity, effective irradiation area of the device and elementary charge (1.6 × 10^−19^ C), respectively. Under the illumination of 0.5 W·m^−2^ light intensity, the detector responsivity R is 3.74 A·W^−1^ and the specific detectivity D* is 3.7 × 10^11^ Jones. Figure 4c demonstrates the trends in the responsivity and specific detectivity in different light intensities. As the light intensity increases, the responsivity and specific detectivity will gradually decrease. The reason for this change trend can be attributed to the photogenic carrier recombination. We have made a detailed analysis in the Appendix A. Since all the SiNWs are regularly distributed on the upper surface of the detector, the detector can sense natural light from different directions. As shown in Figure 4d, the results indicate that the detector can exhibit a good light response to different angles of illumination. In the same light intensity environment, the detection signal of large-angle incident light is stronger. This is an interesting phenomenon. We analyzed that the possible reason is that the large-angle incident light can reflect at the bottom of the groove and then irradiate the SiNWs again We conducted a qualitative analysis in the Appendix A. In Table 1, we compared the SiNW array natural light detector with other types of natural light detectors, the comparison results showed that the SiNW array natural light detector has excellent optoelectronic performance.

Due to the highly active surface of SiNWs, the signal of the detector is difficult to stabilize during the testing process. SiNWs are susceptible to reacting with external substances, which leads to the poor long-term stability of the detector. We design a fully wrapped structure, in which the SiNWs are encapsulated by silicon nitride and silicon oxide. This method prevents the surface morphology of SiNWs from changing over time, greatly improving the stability of test signals and the long-term stability of detectors. As shown in Figure 5a, we removed the oxide layer of a detector and exposed the SiNWs to air. The experiment showed that the electrical characteristics of this detector changed over time in the same light intensity environment, and this comparative experiment proves the improvement effect of the fully wrapped structure on the long-term stability of the detector. Figure 5b shows the comparison of tests before and after 60 days, demonstrating the good repeatability and reliability of the detector.

Through the above structural innovation design, we have effectively improved the response time, signal strength, signal-to-noise ratio and long-term stability of the detector. The high-strength and highly reliable initial signal reduces the cost and complexity of the backend test circuit (there is no need to add circuit modules such as amplification, noise reduction, filtering, etc.). We only used the STM32 microcontroller to process the test signals of the detector, and we built a high-performance SiNW natural optoelectronic detection system. Figure 5c shows the entire calculation and processing flow of the microcontroller signal processing circuit. The testing system includes an SiNW array photosensitive module, an STM32 microcontroller AD conversion module, a memory module and an OLED display module. It is worth mentioning that the circuit structure in the photosensitive module is different from common resistive sensors. We found that the signal at the AO port is interfered by alternating pulse ripples; therefore, we connected an electrolytic capacitor in parallel to the detector, effectively solving this problem.

Before detecting the unknown natural light intensity, we measured the current signal of the detector under different natural light intensities to obtain the standard curve (Figure 4b). We stored these calibrated resistance data and corresponding light intensity values in the main program of the microcontroller. These data will be used as a reference standard for subsequent measurements to ensure the detector can accurately measure the unknown natural light intensity. Then, we wrote and debugged the subroutine code for the AD conversion and OLED display modules. On the other hand, we needed to ensure that it could accurately convert the photoelectric analog signal to a digital signal, which included setting the appropriate conversion mode, channel and AD sampling time. By calling these subroutines from the main program, the system can measure the light intensity signal of the external environment and display the test results in real time on the electronic screen.

After completing the preliminary preparation work, we conducted actual tests in outdoor environment (Figure 5e). When the SiNW array photosensitive module is exposed to sunlight, the increase in carrier concentration leads to a decrease in the detector’s resistance and a decrease in the potential at the AO port. At this point, the corresponding photoelectric signal is measured. Subsequently, the photoelectric analog signal enters the STM32 microcontroller through the GPIO port. By setting appropriate conversion modes, conversion channels and AD sampling times, a 12 bit successive approximation analog-to-digital converter can convert analog signals into digital signals. Then, the photoelectric digital signal is input into the microcontroller memory to obtain the corresponding light intensity value. Finally, the test results are transmitted to the OLED screen for display. As shown in Figure 5e, the test results of the SiNW natural light testing system is the same as the standard equipment in the same light intensity environment, and the testing accuracy of the system reaches ±0.1 W·m^−2^.

## 4. Conclusions

In this article, we achieved a low-cost, highly controllable and large-scale preparation of SiNW array natural light detectors by using traditional microfabrication techniques. In addition, we designed a fully wrapped structure for SiNWs, which protected the fragile SiNWs and improved the long-term stability of the detector. The test results show that the natural light detector has an excellent photoresponse performance. The light responsivity R of the detector is 3.74 A/W, the response time is about 0.015 s, the specific detectivity D* is 3.7 × 10^11^ Jones, and the power consumption is only 0.91 μW. By innovating the structure of the detector, we effectively improved the response time, signal strength, signal-to-noise ratio and long-term stability of the detector and significantly reduced the cost and complexity of the backend test circuit. The initial photoelectric signal can be processed using a low-cost STM32 microcontroller, with a testing accuracy of ±0.1 W·m^−2^. The SiNW array natural light detector in this article has the characteristic of miniaturization, low cost, mass production and high performance, so it has broad application potential in multiple scenarios.

## Figures and Tables

**Figure 1 micromachines-15-01201-f001:**
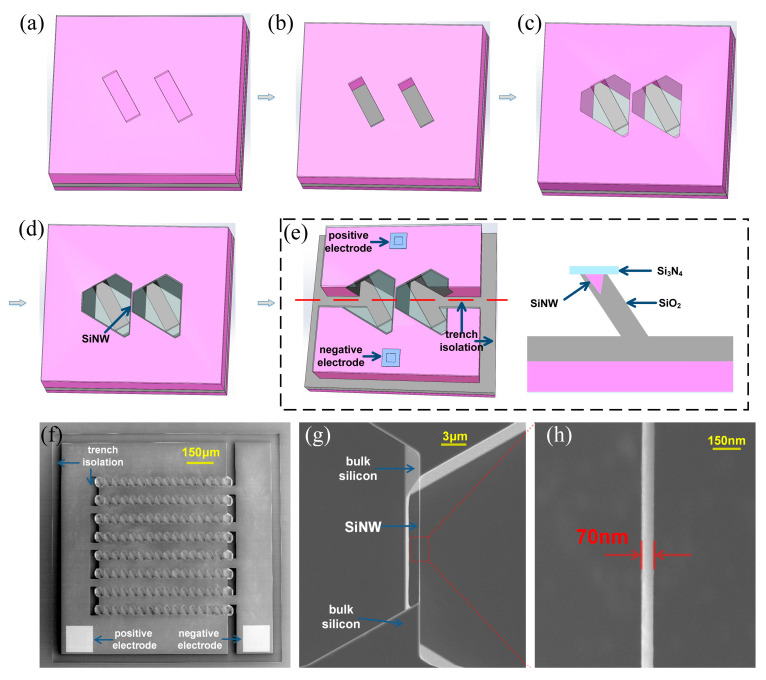
The fabrication process of a single SiNW. (**a**) Transfer rectangular patterns using photolithography technology; (**b**) the rectangular etching grooves are formed by dry etching; (**c**) the preparation of silicon wall structure by anisotropic wet corrosion; (**d**) the SiNW are prepared by using a self-limiting oxidation process; (**e**) the three-dimensional model and cross-section of a single SiNW detection unit; (**f**) the SEM images of SiNW arrays; (**g**) the SEM images of a completely suspended SiNW; (**h**) the localized magnification of the SiNW.

**Figure 2 micromachines-15-01201-f002:**
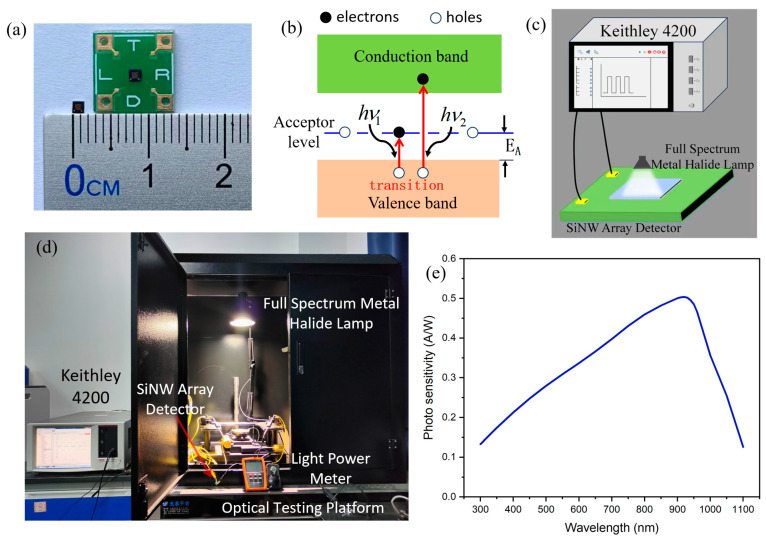
(**a**) The SiNW array natural light sensor chip and its packaged detector; (**b**) the sensing mechanism; (**c**) the test model diagram; (**d**) the optical testing system; (**e**) the photo sensitivity of the SiNW array natural light detector.

**Figure 3 micromachines-15-01201-f003:**
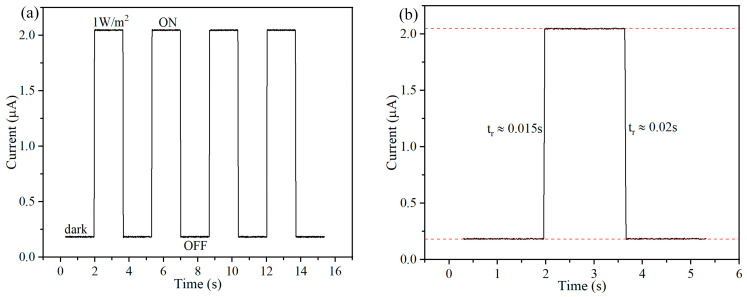
The photoelectric characterization of the SiNW array natural light detector. (**a**) The current signal of the detector in a dark environment and a 1 W·m^−2^ light intensity environment; (**b**) the local magnification of the current signal.

**Figure 4 micromachines-15-01201-f004:**
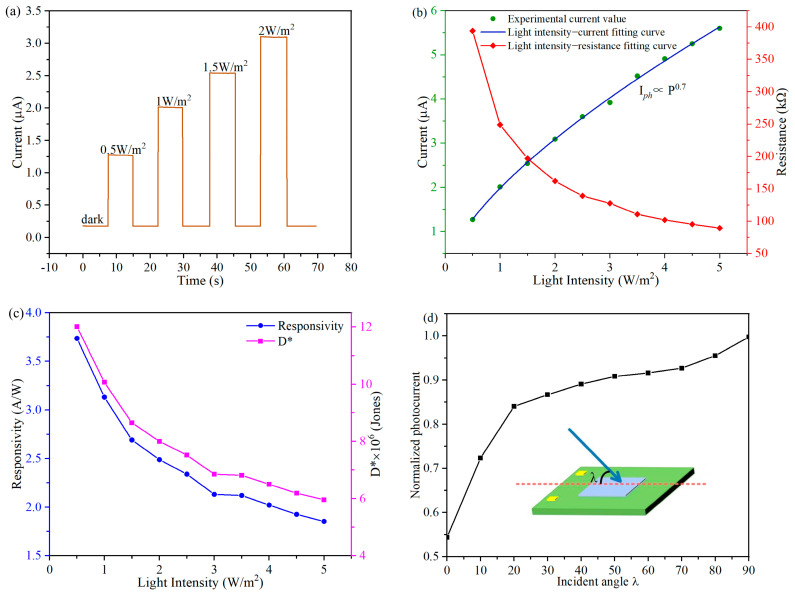
(**a**) The response characteristics of the detector under different light intensities; (**b**) the light intensity–current fitting curve and the light intensity–resistance fitting curve; (**c**) the trends in the responsivity and specific detectivity under different light intensities; (**d**) the response of the detector to incident light at different angles.

**Figure 5 micromachines-15-01201-f005:**
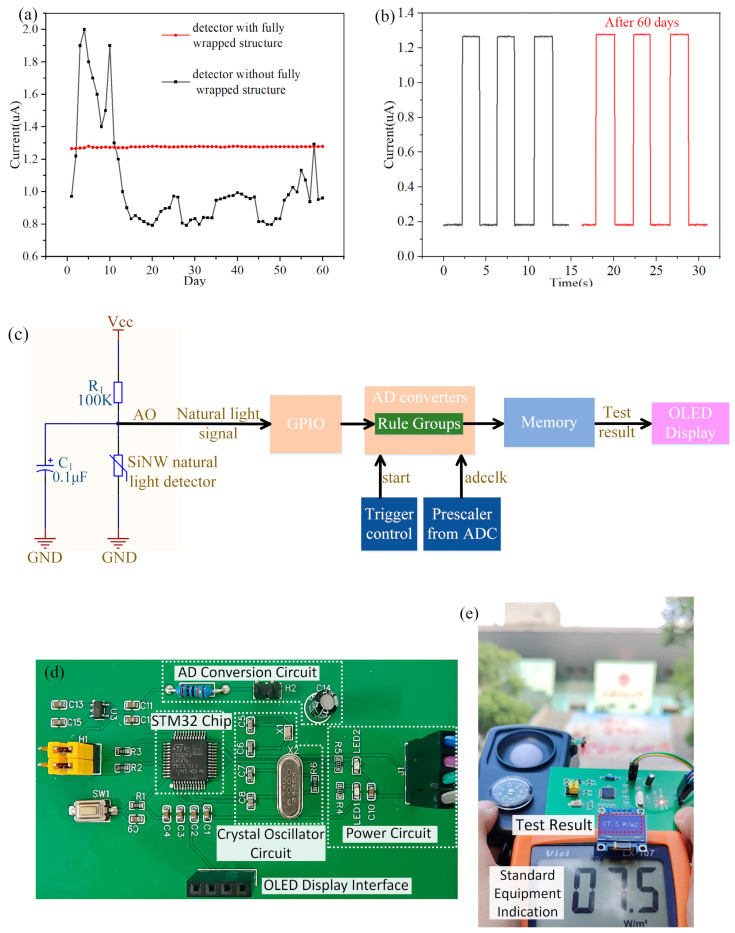
(**a**) The long-term stability testing of two types of detectors; (**b**) the comparison of tests before and after 60 days; (**c**) the block diagram of the SiNW array photosensitive module and microcontroller circuits; (**d**) the microcontroller system PCB board; (**e**) the outdoor natural light test of the SiNW natural light detector.

**Table 1 micromachines-15-01201-t001:** The comparison of the SiNW array natural light detector and other types of natural light detectors.

Material (s)	Bias (V)	Responsivity (A/W)	t_r_/t_d_ (s)	D* (Jones)	Ref.
SiNWs	0.5	3.74	~0.015/~0.02	3.7 × 10^11^	This Work
TiO_2_ NWs	10	~3.5	2.5/2	-	[25]
In_2_Se_3_ NWs	3	~89	<0.3/<0.3	-	[26]
C_60_ NWs	-	-	0.06/0.05	-	[27]
Porous perovskite NWs	1	-	0.12/0.08	-	[28]
TllnSSe single crystal	20	0.61	0.31/0.3	6.24 × 10^11^	[29]
Sb_2_S_3_ nanowall arrays	0.5	-	0.52/1.11	-	[30]

## Data Availability

The original contributions presented in the study are included in the article, and further inquiries can be directed to the corresponding author.

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
