# Peer review of "CMOS-Compatible High-Performance Silicon Nanowire Array Natural Light Electronic Detection System"

_micromachines, 2024, doi:10.3390/mi15101201_

Round 1

Reviewer 1 Report

Comments and Suggestions for Authors

The authors describe a photodetector based on silicon nanowire arrays, and characterize its performance under several light intensities. They also present the device packaged with electronics and a display, and demonstrate its operation in the field.

While I find the manuscript generally satisfactory as a technical report of a photodetector, the context and motivation are sorely lacking as a report of original research. This reviewer believes that this is a major issue that must be thoroughly addressed before this manuscript is suitable for publication in a research journal.

- There are many photodetectors commonly available in the market. The authors have chosen to describe this -- somewhat narrowly -- as a natural light sensor. Why? What distinguishes a natural light sensor from a regular photodetector? Are there certain aspects of the "natural light" application where this device has advantages over conventional products?

- Suppose we only consider just natural light detectors, the authors have not explained why -- even in principle -- silicon nanowires are supposed to be advantageous to regular (unstructured?) silicon photodetectors, except simply using descriptions like "novel", "low-cost", "controllable" without appropriate elaboration or citations.

- The authors have chosen to benchmark against other nanowire or other novel detectors still in a relatively immature stage of development. I believe they need to benchmark against conventional detectors. I can quite easily find commercially available ambient light sensor chips on the websites of major electronics retailers. This brings into question the claims of nanowire advantages that the authors mention:

-- Size: a nanowire is very small, but the authors still have to use an array. How large is this array? The manuscript does not describe adequately. If you have to use a large array, where is the advantage?

-- Cost: The authors quote RMB20, while current products are available for below US$1. It is not clear where the cost scaling advantage comes from.

-- Speed: 20ms response isn't particularly fast, and is probably limited by the control electronics scheme anyway.

-- Current / power consumption: Commercial chips also operate at ~uA levels, with ~uW power consumption. Are nanowires supposed to be better than regular photodiodes in this aspect, and if so, why?

- Why is there a need to use both SiN and SiO2 for device packaging? Is there doping involved in the silicon fabrication? The description here could be clearer.

- The authors report a responsivity of >3000A/W, but this appears to be a value for the amplified photocurrent. If it is amplified, it is not clear how this value is useful for describing and comparing the performance of the base device.

- The authors describe testing the "dynamic response", but in reality is just stepping through a few static measurements rather slowly. This goes against the usual intuition of what "dynamic" means.

- How does the "regular distribution" of nanowires allow the detector to sense light from different directions? Is this not more affected by the orientation of the nanowires? In any case, how does the directionality of the nanowries compare to regular photodiodes?

- The comparision of response curves after 1000 cycles does not seem very meaningful, especially if each cycle is only a few seconds. If the authors really did measure the device for 60 days, they should show a close-up comparison of the device response at the start and end to see how much is has changed or drifted.

Comments on the Quality of English Language

Could cut down on unnecessary repeats. Numerous sentences could be more clearly phrased, but it is not in my scope to provide a detailed review of this. This is best left to a professional English editor.

Reviewer 2 Report

Comments and Suggestions for Authors

The manuscript proposes a new natural light detector based on high-performance silicon nanowire (SiNW) arrays. The proposed solution is implemented as a prototype and experimentally evaluated, demonstrating its performance. The manuscript has good merits and is suitable for publication. Please find below my detailed comments.

1.       The abstract of the manuscript presents the accomplishments of the article in an adequate manner, clearly emphasizing what has been made, how, and which are the results. I would recommend the authors to slightly restructure the length of the phrases, which are too long. Breaking the long phrases in smaller ones would improve the presentation stile. Example : “In this article, we propose a novel natural light detector based on high-performance silicon nanowire (SiNW) arrays, we achieved highly controllable, miniaturized, arrayed, low-cost and large-volume fabrication of SiNW natural light detectors by using only a conventional micromachined CMOS process.” Nevertheless, many of the phrases in the abstract are similar, requiring improvement.

2.       The observation above is applicable for the entire manuscript. I would recommend the authors to carefully read the manuscript and to try to divide the extra-long phrases in to shorter ones. Using extra-long phrases by non-native English speaking persons makes the manuscript slightly difficult to fallow. See another example: “Figure 2(a) shows the SiNW array natural light sensor chip and its packaged device, and Figure 2(b) shows the experimental test system, we used a full spectrum metal halide lamp as the light source, which is a common sunlight simulator, the Puyuan DP832 DC voltage source supplies power to the detector, we use the LX-107 optical power meter as the standard equipment to calibrate the optical power density of the light source.” Numerous ideas are forcedly connected in a single phrase.

3.       As I mentioned above, the abstract is very good, but I would add a short sentence mentioning the purpose of this work, describing which is the context, or how can this novel natural light detector can be used (in what kind of application).

4.       The introduction adequately presents the context of the work and provides a short state of the art regarding other light sensing devices. Although rather short, the state of the art is clear and related to the topic of the manuscript.

5.       Section 3 briefly presents the SiNW array fabrication process, providing the necessary details.

6.       Section 4 provides the details concerning the testing setup, testing procedure and experimental results. I found this section to be sufficiently detailed and presented in an adequate manner. The experimental testing procedure is adequate, whereas the experimental results are convincing. The provided figures are of good quality. Moreover, the results achieved are compared with those from other works, further emphasizing their value.

In conclusion, in my opinion, the manuscript is suitable for publication. The experimental implementation and the experimental testing confirming the concept performance are the strong point of the manuscript. Although the presentation stile must be improved for better understanding, the manuscript is valuable and suitable for publication. The authors must carefully optimize the presentation style for better understanding, and to avoid the extra-long phrases. The authors use extra-long phrases, with ideas that are forcedly connected with each other. These long phrases must be divided in shorter sentences which are connected with each other in a logical manner.

Comments on the Quality of English Language

The presentation of the manuscript must be improved. The authors use extra-long phrases, with ideas that are forcedly connected with each other. These long phrases must be divided in shorter sentences which are connected with each other in a logical manner.

Reviewer 3 Report

Comments and Suggestions for Authors

Authors demonstrate a suspended SiNW array to apply for white light detection with significant novel performances in comparaison with other types of phodetectors. The introduction to the technology is clearly developed as well as the device characterization. However I note to consider the following remarks and questions : 

- It is important to include values of the spectral absorption and spectral sensitivity of the SiNW detectors.       

- Also give a clear view of the ON - OFF time response, is there a transient contribution in fig 3 ; How to reduce and control the response time measured as 0.04 s  for 1W/m² ?                                                                         -The detectivity decreases as a function of light intensity in fig 4. Precise the physical mechanisms which are concerned ? increase due to the dark current contribution ? 

 - The response to large incident light angles in fig 4 may be influenced by beam polarization and light beam reflection on the interface ?  

To conclude the obtained results relevant to the technology and performances of the novel SiNW detector array are well suited for publication in the journal, but the authors must take account of the above question and comments in the final form of the manuscript. 

Round 2

Reviewer 1 Report

Comments and Suggestions for Authors

In the previous version, I commented that the context and motivation are sorely lacking as a report of original research. I must say that the authors' response has been thoroughly unsatisfactory, especially in the area of comparison to regular photodetectors.

1) Despite having made the point that benchmarking is essential, the authors have only made a few remarks in the point-by-point response, but did not try to include any of that in the main manuscript. This is very disappointing. 

2)  I am not saying the authors have to include another benchmarking table in the manuscript, but they can survey a representative number of them, make some conclusions or generalizations, and cite accordingly.

3) Saying that photodetectors "focus on a specific wavelength" is simply not appropriate nor correct. Any regular silicon photodetector should have the same inherent spectral response as your nanowire, unless the nanosized structure, in itself, modifies the spectral properties in some way. In other words, the authors have not shown that there is an inherent difference between the two, and are better off not trying to say that they are different.

4) The point of whether nanowires -- in principle -- have any advantage over regular photodetectors is not addressed beyond simply one phrase "due to the slender morphology of SiNWs". This is simply inadequate. If the advantage is not obvious, and the authors are exploring to see if such an advantage exists, then they should simply say so directly. 

5) The authors did not address the question of where the advantage of nanowires is (against regular photodetector) if you have to group them into arrays (1.3 x 1.3 mm) to have better performance.

5) The use of the LS06-S as a "benchmark" is inappropriate, and close to being dishonest. Surely the authors are not suggesting that a bulky product with two lead pins are "widely used in mobile phones"? 

6) The point of benchmarking is to survey the landscape and compare widely. The authors cannot claim a lack of readily available commercial ambient light sensors to compare against. Yet they only highlight one example which happens to be worse in working current (which is related to the power consumption)

7) Literally the first product that came up when I searched for ambient light detector is the Texas instruments OPT3004. It has a working current of 1.8uA (similar to authors' device), and sensor size of 0.5 x 0.4mm (even smaller than the authors' device!), and costs US$1. There are many more products out there. Again, without rigorous benchmarking, the authors cannot claim the nanowires are better.

8) The authors claim the repsonsivity value of 3000A/W is not amplified. This must be a mistake. Responsivity = current / power. Current is ~10^-6 A. Power is intensity (1W/m^2) * area (~10^-3m * 10^-3m) ~ 10^-6 W. So R ~ 1 A/W, by the authors' own data.

Comments on the Quality of English Language

Same as previous comment; I recommend a professional English editor.

Reviewer 3 Report

Comments and Suggestions for Authors

The authors provided very relevant answers and comments to the questions which contribute to the quality of the manuscript. The revised form is now well suited for publication in the journal.